# Chemical Methods for Microbiological Control of Winemaking: An Overview of Current and Future Applications

Francesco Tedesco [1], Gabriella Siesto [1,*], Rocchina Pietrafesa [1], Patrizia Romano [2], Rosanna Salvia [3], Carmen Scieuzo [3], Patrizia Falabella [3] and Angela Capece [1]

1 Scuola di Scienze Agrarie, Forestali, Alimentari ed Ambientali, Università degli Studi della Basilicata, Via dell'Ateneo Lucano 10, 85100 Potenza, Italy
2 Dipartimento di Economia, Universitas Mercatorum, 00186 Roma, Italy
3 Dipartimento di Scienze, Università degli Studi della Basilicata, Via dell'Ateneo Lucano 10, 85100 Potenza, Italy
* Correspondence: gasiesto1@virgilio.it; Tel.: +39-0971205585

**Abstract:** Preservation technologies for winemaking have relied mainly on the addition of sulfur dioxide ($SO_2$), in consequence of the large spectrum of action of this compound, linked to the control of undesirable microorganisms and the prevention of oxidative phenomena. However, its potential negative effects on consumer health have addressed the interest of the international research on alternative treatments to substitute or minimize the $SO_2$ content in grape must and wine. This review is aimed at analyzing chemical methods, both traditional and innovative, useful for the microbiological stabilization of wine. After a preliminary description of the antimicrobial and technological properties of $SO_2$, the additive traditionally used during wine production, the effects of the addition (in must and wine) of other compounds officially permitted in winemaking, such as sorbic acid, dimethyl dicarbonate (DMDC), lysozyme and chitosan, are discussed and evaluated. Furthermore, other substances showing antimicrobial properties, for which the use for wine microbiological stabilization is not yet permitted in EU, are investigated. Even if these treatments exhibit a good efficacy, a single compound able to completely replace $SO_2$ is not currently available, but a combination of different procedures might be useful to reduce the sulfite content in wine. Among the strategies proposed, particular interest is directed towards the use of insect-based chitosan as a reliable alternative to $SO_2$, mainly due to its low environmental impact. The production of wines containing low sulfite levels by using pro-environmental practices can meet both the consumers' expectations, who are even more interested in the healthy traits of foods, and wine-producers' needs, who are interested in the use of sustainable practices to promote the profile of their brand.

**Keywords:** sulfur dioxide ($SO_2$); chemical methods; antimicrobial activity; microbiological control; chitosan; sustainable approaches; winemaking

## 1. Introduction

Wine fermentation is a microbiologically complex process which requires the monitoring of the microorganism load at different stages of the process. Winemakers manage the different steps of this process with the aim to obtain high-quality products according to their wishes and the expectations of their customers. However, the chemical characteristics and microbial composition of wine are in constant evolution throughout the process, and some parameters are difficult to control.

Microbial metabolism is one of the multiple factors affecting wine quality, by contributing to its complexity or, in some cases, leading to undesirable aromas. The control of wild microorganisms present in grape must is an advisable oenological practice to ensure the imposition of yeast starter cultures for adequate alcoholic fermentation, whereas the

control of microorganisms present in the wine is necessary both to assure the dominance of suitable bacteria strains for malolactic fermentation (MLF) [1], and to avoid wine spoilage due to the growth of undesirable yeasts, lactic acid bacteria (LAB), and acetic acid bacteria (AAB), with irreversible effects on wine quality and considerable economic losses. Some LAB and AAB species, such as *Lactobacillus* spp., *Pediococcus* spp., and *Acetobacter* spp., are primarily responsible for the loss of quality of musts and wines, in consequence of the formation of undesirable aroma and flavor compounds leading to defects such as a "vinegary," "nail polish-remover" taste. The major spoilage yeasts include species and strains of the genera, *Brettanomyces*, *Candida*, *Hanseniaspora*, *Pichia*, *Zygosaccharomyces*; in particular, yeasts belonging to the genus, *Dekkera* (anamorph *Brettanomyces*), are producers of 4-ethylphenol, 4-ethylguaiacol, and tetrahydropiridine, which are considered off-flavors in wine. To date, preservation technologies for winemaking have relied mainly on the addition of sulfur dioxide ($SO_2$), in consequence of the large spectrum of action of this compound, linked to the control of undesirable microorganisms and the prevention of oxidative phenomena [2].

However, in a context of societal concern regarding food and wine preservation, reducing the sulfite level in the wine now represents a major challenge for the wine industry. Different factors, such as increasing consumers' attention toward health concerns, the potential organoleptic alterations of the final product, and restrictive legislation on preservatives [3], along with the quest for environmentally friendly production, have driven the interest of scientific community and producers toward alternative methods to $SO_2$ [4]. Furthermore, the massive employment of $SO_2$ is not always compatible with the production of high-quality wines, as it can cause organoleptic alterations in the final product, neutralize the aroma, and even produce characteristic aroma defects, such as undesirable aromas of the sulfurous gas, when this compound is reduced to hydrosulfate and mercaptanes. In addition, the use of excessive $SO_2$ doses does not always avoid the risk of wine spoilage, in consequence of the emergence of tolerant/resistant spoilage microorganisms [5].

This paper is a review summarizing the main findings regarding chemical methods that have been so far studied to substitute or reduce the use of $SO_2$ in winemaking. After a summary on the main characteristics of $SO_2$ application in wine, the action mechanisms of the different methods will be discussed, including their efficacy, drawbacks, and effects on the final quality of the wine in order to claim that the new alternatives apply the same $SO_2$ properties in wine.

Special emphasis will be placed on very innovative methods, such as the use of chitosan extracted by unconventional sources, to develop an integrated perspective on how to produce a more natural, healthier, and sustainable wine.

## 2. Sulfur Dioxide

Sulfur dioxide ($SO_2$) is an additive commonly used for the microbiological control of foods, in particular for acidic foods, such as fruit juice and wine [6]. In the winemaking process, the wide use of this compound is correlated to its large spectrum of action, mainly linked to the control of undesirable microorganisms and the oxidative processes, as summarized in Table 1.

$SO_2$ is added to grape must or wine as a liquid or gaseous form; in Europe and Switzerland, the $SO_2$ forms which can be used as food additives are sulfur dioxide, sodium sulfite, sodium hydrogen sulfite, sodium metabisulfite, potassium metabisulfite, calcium sulfite, calcium hydrogen sulfite, and potassium hydrogen sulfite [6].

**Table 1.** Effects of $SO_2$ addition in grapes, must, and wine.

| Process Phase | Times of $SO_2$ Addition | $SO_2$ Action |
|---|---|---|
| Grape and Must | Before the start of alcoholic fermentation | - Antimicrobial activity towards non-*Saccharomyces* yeasts, acetic and lactic acid bacteria (in white winemaking to avoid the malolactic fermentation) [7–9]<br>- Antioxidant activity to avoid the must oxidation [2,10] |
| Wine | Filtration, decanting, and aging | - Antimicrobial activity towards *Brettanomyces* spp., *Candida* spp., *Pichia* spp., acetic acid bacteria [11–14]<br>- Antioxidant activity to avoid the wine oxidation [11,15] |
| | Before bottling | - Antioxidant activity to avoid the wine oxidation [16–18]<br>- In sweet wines to avoid the refermentation process by Saccharomyces cerevisiae [19,20] |

Depending on the pH, different $SO_2$ forms can be found in the wine, which are sulfurous acid, or molecular $SO_2$, ($H_2SO_3$), bisulfite ion ($HSO_3^-$), and sulfite ion ($SO_3^{2-}$). At wine pH (ranging between 3 and 4), $HSO_3^-$ is the predominant form, whereas the sulfite ion is negligible [21], as this form is found at pH > 7; the non-dissociated fraction, $H_2SO_3$, is commonly found at pH < 2. These three compounds form the "free $SO_2$", but the complex chemical equilibrium of the $SO_2$ in the wine results in different $SO_2$-combined compounds. Indeed, bisulfite ion can bind some wine compounds, such as acetaldehyde, glucose, quinones, anthocyanins, and ketoacids, forming compounds more or less active against microbiological spoilage. The stable combination with acetaldehyde makes $SO_2$ very poorly available for wine protection, conversely to the combination with other compounds species, which is reversible.

$SO_2$ is present in wine not only as a consequence of exogenous addition during the winemaking process, but it is also produced by yeast metabolism during alcoholic fermentation, the so-called "biological $SO_2$". In fact, yeasts use the sulfur present in the must for the synthesis of amino acids, and the production levels of sulfites by yeasts is highly strain-dependent [22].

As reported by Zara and Nardi [23], sulfate ($SO_4^{2-}$) in must is transported into the *Saccharomyces* yeast cell, where $SO_4^{2-}$ is reduced to sulfite ($SO_3^{2-}$) and then into sulfide ($S^{2-}$). $SO_3^{2-}$ is transported outside the cell, and $S^{2-}$ is incorporated in methionine and cysteine, whereas the sulfide in excess diffuses outside the cell (Figure 1). At the end of alcoholic fermentation, yeasts produce sulfites and sulfides. An excessive quantity of these compounds can cause problems in finished wine because sulfides can lead to off-flavors (e.g., rotten eggs), and sulfites at a high concentration can delay the onset of MLF by inhibiting LAB [22].

In consequence of these metabolic activities, yeasts produce $SO_2$ both in must and finished wine, and, based on the production level of $SO_2$ quantity, the yeasts are divided into low (<10 mg/L), middle (10–30 mg/L), and high (>30 mg/L) $SO_2$-producers.

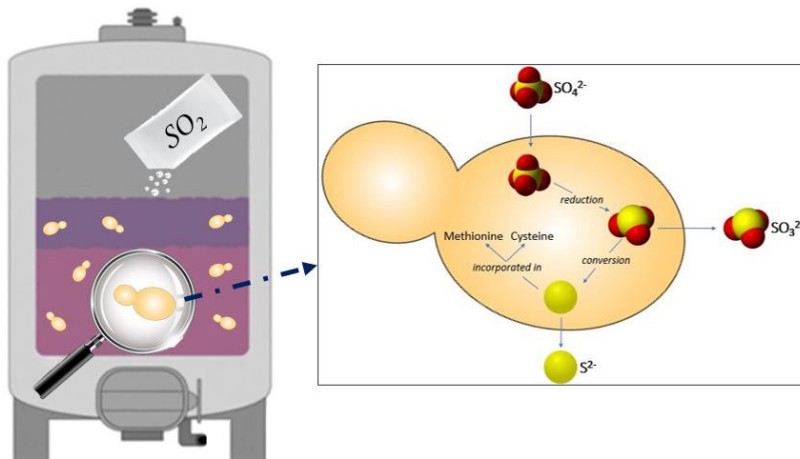

**Figure 1.** Sulfur metabolism in *Saccharomyces cerevisiae* yeast cell.

### 2.1. Antiseptic Activity of SO$_2$

As regards the antiseptic activity of SO$_2$, this compound inhibits the development of different microorganisms, with the highest antimicrobial activity against bacteria (LAB and AAB), followed by non-*Saccharomyces* yeasts, whereas usually *Saccharomyces* yeasts are highly resistant, although the sensitivity level is strain-specific [7].

The highest antimicrobial activity is shown by molecular SO$_2$, whereas bisulfite ions show weak antimicrobial activity, and the bound SO$_2$ exerts only a low antibacterial action.

#### 2.1.1. Activity against Yeasts

As regards the antimicrobial activity against yeasts, the addition of SO$_2$ in the grape must is mainly to inhibit the growth of non-*Saccharomyces* yeast species prevalent at this stage. It was reported that, in spontaneous wine fermentation, concentrations of total SO$_2$ higher than 40 mg/L compromise the development of most of these yeasts, with the exception of *Hanseniaspora osmophila* and *Candida* spp. [24], whereas concentrations below 40 mg/L inhibit the development of *Metschnikowia* spp., and concentrations between 60 and 100 mg/L inhibit the development of *Torulaspora* spp. [25], *Zygosaccharomyces bailii*, *Schizosaccharomyces pombe*, and *Saccharomycodes ludwigii* have been described as highly tolerant species to SO$_2$ [26,27], whereas *Kloeckera apiculata* and *Hansenula anomala* are highly sensitive to this compound [28].

The addition of SO$_2$ in the final wine is attributed mainly to the control of yeasts belonging to the genus, *Dekkera/Brettanomyces*. These are very harmful yeasts for the production of wine, as they are able to resist environmental conditions poor in nutrients, colonize and contaminate cellar equipment (especially wooden barrels), and produce high amounts of undesirable compounds, such as acetic acid, ethylphenols, and tetrahydropyridines [29,30].

Several studies have monitored the development of these contaminant yeasts in wine, showing the complexity in preventing the development of *Brettanomyces* spp. Survival models of *Brettanomyces bruxellensis* after the addition of sulfur dioxide [31] showed that the control of these yeasts can be obtained by adding relatively high doses of the antimicrobial compound (approximately 1 mg/L of the active molecular fraction). Furthermore, these authors confirmed previous findings [11], indicating that the effective control of *B. bruxellenis* should also be carried out by avoiding, as far as possible, the contact with oxygen, which favors the reactivation of yeast cells in the viable, but not culturable (VBNC), state. The maintenance of a free sulfur dioxide dose of 25–35 mg/L seems to be effective in eliminating viable *B. bruxellensis* cells [6].

The effect of sulfur dioxide on microbial cells has been studied for several decades, and mostly in the yeast, *Saccharomyces cerevisiae*. The molecular SO$_2$ enters the yeast cell through the plasma membrane with a simple diffusion mechanism [32]. Once inside the

cell, in consequence of the higher intracellular pH (about 6.5), molecular $SO_2$ is largely converted to $HSO_3^-$ [33].

However, in the yeast cell, the dissociation of $H_2SO_3$ reduces the intracellular concentration of molecular $SO_2$, allowing further diffusion into the cells until the $SO_2$ concentration is balanced on both sides of the plasma membrane [34]. $SO_2$ is a highly reactive molecule, which it binds to many molecules in the cell (proteins, nucleic acids, coenzymes, cofactors, vitamins, etc.), interfering with intracellular processes [35]. This mechanism inhibits microbial growth.

The general tolerance to $SO_2$ exhibited by *S. cerevisiae* is correlated to detoxification mechanisms developed by this yeast. The main mechanisms developed by the yeast cell to resist sulfites are the following [36]:

- Metabolic pathways involved in the production of sulfur compounds, such as the amino acids methionine and cysteine [37];
- Sulfite detoxification by the membrane efflux [37];
- Production of $SO_2$-binding molecules, such as acetaldehyde [23];
- The cell entry into a VBNC state [38]. This physiological state is described as a protection strategy in which the cells can wait for more favorable conditions. In wine, the VBNC state allows spoilage yeasts and bacteria to survive throughout the wine fermentation process and into the wine bottle. In wine conditions, the presence of chemical stressors such as $SO_2$ has been shown to induce a VBNC state in *S. cerevisiae* and other yeast species, such as *Brettanomyces/Dekkera bruxellensis*, and reactivate later, when conditions become more favorable [39]. The removal of $SO_2$ from the wine environment can be obtained by increasing the pH in order to shift the chemical equilibrium of $SO_2$ with a decrease of the concentration in molecular $SO_2$, favoring the exit from the VBNC state [40].

### 2.1.2. Activity against Bacteria

The $SO_2$ provides, in addition to the control of undesirable yeasts, an antibacterial activity both in must and wine, although the action mechanism has not been clarified yet. Conversely to yeasts, the bacteria are weakly inhibited by bound $SO_2$, which exerts an antibacterial activity from 5 to 10 times lower than free $SO_2$, but it should be considered that bound $SO_2$ can be from 5 to 10 times more abundant in comparison to the free form.

The $SO_2$ antibacterial activity is mainly attributed to AAB and LAB.

As regards the AAB, the predominant species on the grapes, especially on berries with a non-optimal health state, are strains of the genera, *Gluconobacter*, *Acetobacter*, *Gluconacetobacter*, and *Komagataeibacter* [41]. The proliferation of AAB, especially those of the *Acetobacter* genus, leads to oxidative phenomena, with undesirable consequences for wine quality. For this reason, the control of AAB is carried out by the use of $SO_2$ and by the control of oxygen, avoiding the contact of wine with oxygen [41].

As with *Brettanomyces* spp., AAB are also able to survive in the VBNC state when $SO_2$ is added and the oxygen is removed, maintaining their metabolic activity and resuming their development when the conditions permit [42]. Indeed, it was reported [11] that *Acetobacter pasteurianus* is able to survive in VBNC under anaerobic conditions and in the presence of $SO_2$, whereas it resumes the viable state with oxygen addition.

The most frequent LAB in wine are mainly species belonging to the genera, *Oenococcus*, *Pediococcus*, and *Lactobacillus* [6].

*Oenococcus oeni*, the LAB mainly responsible for MLF in wine, is able to overcome the stressful conditions of wine (low pH, high ethanol content), but it has a high sensitivity to $SO_2$ [43]. The sensitivity of *O. oeni* to $SO_2$ is a positive trait in the production process of white wines, where, usually, MLF is not desired, whereas for red wines, MLF is favored in part by the survival of the bacterium in VBNC, and in part by the inoculation of starter cultures that promote rapid bioconversion [8].

## 2.2. Technological Activities of SO$_2$

Other than the antimicrobial action, the addition of sulfur dioxide to wine has further activities. This compound is highly effective in preventing chemical and enzymatic oxidative processes in winemaking. Chemical, or non-enzymatic, oxidation processes may occur in grapes, must, and wine, and can cause changes in color and sensory characteristics. During this process, reactive oxygen species, produced by reduced transition metals ions (e.g., Fe (II), Cu (I)), react with many constituents of wine, especially phenolic compounds, causing oxidation [15]. These reactions lead to the formation of quinones, unstable compounds that can undergo further reactions, with the final formation of pigments, responsible for wine color alteration [44]. As an antioxidant, SO$_2$ is able to react with the oxygen-reduced form to inhibit aldehyde formation and to reduce quinones back to their phenol form [45].

The SO$_2$ addition, especially in the form of HSO$_3^-$, is fundamental to inhibit enzymatic oxidation processes [21], which occur in grapes and must as a consequence of the activity of polyphenol oxidase (PPO) enzymes, such as tyrosinase from grapes and laccase produced by *Botrytis cinerea*, and peroxidase (POD) from grapes [46]. This enzymatic process occurs rapidly during and after the crushing operation.

## 2.3. Negative Effects of Sulfur Dioxide

Despite the numerous advantages found in the use of SO$_2$ during winemaking, drawbacks in its use have to be considered, both on wine quality and health consumers.

As regards the influence on wine quality, excessive doses can cause the appearance of sensory defects, bad smells, and unpleasant aromas [44]. These off-flavors are related to the formation, starting from the added SO$_2$, of hydrogen sulfide and mercaptans, which is observed in fermentations performed in strict anaerobiosis conditions, and in cases of prolonged contact of the wine with the lees [6].

Furthermore, during alcoholic fermentation in the presence of excessive doses of SO$_2$ and of a must poor in nutrients, the yeasts degrade the sulfur dioxide, with formation of typical off-flavors related to sulfur compounds, such as rotten eggs.

As regards the health effects, the ingestion of sulfites can cause problems, especially in sensitive subjects. The observed reactions can affect the skin (urticaria, angioedema, hives, and pruritus), respiratory system (bronchospasm, bradycardia, etc.), and gastrointestinal apparatus (nausea, stomach cramps, and diarrhea) [47]. These symptoms are not common in all people; adverse reactions are caused in a very small population (about 1%) of "sulfite-sensitive" individuals, most being very mild [48].

In consequence of such evidence, the World Health Organization (WHO) estimated the allowable daily intake of SO$_2$ to be about 0.7 mg per kg of body weight, and the European Community imposed dose limits of SO$_2$ for the different foods. For winemaking, the European Regulation (EU Regulation No. 606/2009 and No. 479/2008) has established the maximum doses of total SO$_2$ contained in wine, which have to be 150 mg/L for red wine and 200 mg/L for white and rosé or pink wines, in cases of wines containing a maximum of 5 g/L of reducing sugars. These doses can be increased by 50 mg/L when the sugar content in the wine (glucose + fructose) exceeds 5 g/L, reaching, 200 mg/L for red wines and 250 mg/L for white wines. Furthermore, specific doses have been defined for some wines, for example, for special wines (Bordeaux superieur, Tokaji, Moscato of Pantelleria, etc.), where the maximum doses of SO$_2$ can reach up to 400 mg/L. The EU Regulation No. 203/2012 establishes the maximum doses for organic wine as 100 mg/L for red wine, and 150 mg/L for white and rosé wines. Moreover, the EU Regulation has obliged to label the wine bottle with the phrase, "contains sulfites", when the SO$_2$ concentration exceeds a quantity of 10 mg/L.

## 3. Alternative Methods to SO$_2$

Although sulfur dioxide is a compound with numerous advantages in winemaking, especially at a microbiological level, consumers' increasing attention towards "healthy"

products free of chemical additives is encouraging research on alternative preservation methods for reducing $SO_2$ use [6]. Furthermore, by considering the wide use of this compound in different food products, the risk is correlated to an excessive cumulative intake, and the World Health Organization (WHO) recommended the reduction of this preservative in sectors in which the use of $SO_2$ significantly contributes to daily intake, and this is the case for wine, particularly where it is regularly consumed.

For this purpose, several compounds have been authorized for the microbiological control of the winemaking process, in order to replace $SO_2$ or limit the used amounts (Table 2).

**Table 2.** Main applications, authorized doses, and antimicrobial activities of chemical compound alternatives to $SO_2$.

| Compound | Chemical Structure | Admitted Amount | Winemaking Stage | Antimicrobial Activity |
|---|---|---|---|---|
| Sorbic acid | | 200 mg/L | Wine storage of sweet wines | Yeasts (*S. cerevisiae*, *Candida* spp.) in association with $SO_2$ [7,49] |
| Lysozyme * | | 500 mg/L (considered as cumulative, taking into account any additions to the must) | - Alcoholic fermentation in white wine<br>- Maceration during alcoholic fermentation in red wine | Gram-positive bacteria (not active against Gram-negative bacteria and the yeast cell) [50] |
| Dimethyl dicarbonate (DMDC) | | 200 mg/L (with no residues in the marketed wine) | Prior to bottling in wine with sugar content ≥5 g/L | Yeasts (*Zygosaccharomyces bailii*, *Zygoascus hellenicus*, and *Lachancea thermotolerans*) [51] |
| Chitosan | | 10 g/hL | - Alcoholic fermentation in all the wines<br>- Red wine aging in barrique | - Non-*Saccharomyces* yeasts, lactic acid bacteria [52,53], *Acetobacter* spp. [54]<br>- *Brettanomyces* spp. [52] |

* = part of the lysozyme chemical structure.

In this review, we will report the current state and recent updating of the main chemical methods actually approved or studied to reduce the sulfite content of the final wine, with special attention to environmentally friendly compounds to improve the sustainability aspects of winemaking.

### 3.1. Sorbic Acid

Sorbic acid is a short-chain unsaturated fatty acid not very soluble in water, but soluble in ethanol (112 g/L at 20 °C), used as an antimicrobial and antifungal in food preservation. In winemaking, it is used as potassium sorbate, which contains 75% sorbic acid and it is soluble in water [7].

The antimicrobial activity of sorbic acid is due to the combination of this molecule with the hydrosulfide group of the microbial enzymatic system, by destroying its activity [55].

The EU Regulation No. 606/2009 imposes a concentration limit of sorbic acid in wine on the market at 200 mg/L. These concentrations are highly tolerated by several contaminant yeasts present during winemaking, such as *Z. bailii* [56], *Brettanomyces* spp. [57], and *Saccharomycodes* spp. [58].

Sorbic acid is also inactive against acetic acid and malolactic bacteria, which are significantly affected by concentrations higher than 0.5–1 g/L [7].

In consequence of these considerations, sorbic acid plays its activity, in association with $SO_2$, against yeasts, such as *S. cerevisiae*, to avoid bottle refermentation during the conservation of sweet wines [49], and against flor yeasts (*Candida* spp.) able to develop on the wine surface [7]. Although the sorbic acid at the imposed concentrations does not modify the organoleptic characteristics of wine, in red wines, the LAB are able to react with sorbic acid, producing unwanted volatile compounds, responsible for a "geranium aroma" [7].

### 3.2. Lysozyme

Lysozyme is a muramidase enzyme isolated from egg white (EC 3.2.1.17) that can be proposed as a substitute of $SO_2$. It is a globular basic protein that consists of a single polypeptide chain of 129 amino acids, characterized by a molecular weight (MW) of about 14.4 kDa with an isoelectric point (pI) of 10.7, and the four disulfide bridges present in this molecule cause a high thermal stability of the enzyme. This enzyme is able to cause the lysis of the bacterial cell, leading to its death [59]; in particular, it breaks the glycosidic bonds between N-acetylmuramic acid and N-glucosamine in the bacterial wall of Gram-positive bacteria. For this reason, lysozyme is active against Gram-positive bacteria, and it is scarcely or not at all active against Gram-negative bacteria (AAB) and the yeast cell [50]. In winemaking, lysozyme is used for the partial replacement of sulfur dioxide in different stages of the technological process, and as a fining agent [60]. In grape must, lysozyme decreases the cell population of LAB [61] without affecting yeast activity and the progress of fermentation. Gram-positive bacteria responsible for must contamination consist of species such as *Pediococcus* spp., *Lactobacillus* spp., *Leuconostoc mesenteroides*, and *O. oeni* [62]. As reported by Delfini et al. [50], based on the composition of their cell wall, they show different degrees of sensitivity or resistance to lysozyme: *O. oeni* is more sensitive than *Lactobacillus* spp. and *Pediococcus* spp., which are resistant to higher concentrations of lysozyme.

The control of LAB is useful in white winemaking, where the addition of 125–250 mg/L of lysozyme allows the inhibition of MLF [62], and in red winemaking, to avoid the increase of volatile acidity and for the optimal regulation of MLF development. In this case, the addition of lysozyme inhibits the bacterial proliferation during prolonged maceration times, avoiding fermentation blocks and facilitating the subsequent inoculation of the selected malolactic starter [61].

As reported by Bartowsky et al. [63], in red wines, lysozyme is poorly effective, as the polyphenolic compounds can inhibit its effect, whereas the best effect is found in white wines. However, the lysozyme can induce heat instability (haze); consequently, clouding phenomena may occur due to protein haze, and a protein stabilization is necessary after the treatment of white wines with lysozyme.

As regards the limits of use, Regulation CE No. 606/2009 requires a maximum addition of 500 mg/L in the must, whereas in wine, this dose must be considered as cumulative, taking into account any additions to the must. Furthermore, by considering that lysozyme used in winemaking is isolated from egg white and it is designated as the egg allergen, Gal4, it might give rise to an allergic reaction in people sensitive to eggs, even in small amounts, and especially in wines not treated with bentonite, which leads to the absorption and precipitation of proteins [64]. In consequence of this, the current European regulation (directive 2007/68/CE) requires that for concentrations equal to or greater than 0.25 mg/L, the presence of lysozyme has to be reported on the label.

The addition of lysozyme, however, does not lead to negative changes in the alcohol content, pH, and organoleptic characteristics of the wine; in red wines, egg proteins containing lysozyme bind tannins by electrostatic interactions, preventing color loss [60]; furthermore, as reported by Sonni et al. [65], the use of lysozyme, together with the addition

of oenological tannins, improves the aroma of wine in consequence of the higher content of esters and acids.

Other studies [66] confirmed that wines obtained with the use of combinations of lysozyme and dimethyl dicarbonate (DMDC), another compound proposed as alternative to $SO_2$ (which will be discussed later), have better aromatic characteristics than wines treated with $SO_2$ alone.

All these considerations indicate that the use of lysozyme might lead to a reduction of $SO_2$, but it cannot be used for total replacement of sulfites alone, since it has only antibacterial activity, but it is not able to control the proliferation of contaminating yeasts and Gram-negative bacteria, nor oxidation phenomena in musts and wines.

### 3.3. Dimethyl Dicarbonate

Dimethyl dicarbonate (DMDC) is an organic compound possessing antimicrobial activity. The European Union, Australia, and the USA have authorized its use in winemaking, and the CE Regulation No. 606/2009 authorizes the use of DMDC up to a maximum dose of 200 mg/L, but without residues found in the wine on the market. Furthermore, the same regulation foresees the addition during the bottling only in wines with a sugar content equal or greater than 5 g/L, whereas in the USA, the DMDC can also be added during wine aging.

In wine, DMDC is quickly hydrolyzed to methanol and carbon dioxide within 12 to 24 h, though the concentration of these compounds does not cause toxicological effects in the wine; however, to ensure the efficacy of this compound, a rapid and adequate homogenization is necessary [51]. The most common commercial application is the addition to wines just prior to bottling [67]. Furthermore, Delfini et al. [68] have shown that it does not alter the organoleptic characteristics of wine.

The action mechanism has been known for some time; this compound reacts with the amino groups of some enzymes in the microbial cell, such as alcohol dehydrogenase and glyceraldehyde-3-phosphate [69], causing the methoxy carbonylation of nucleic residues [67], and leading to the death of the microbial cell.

DMDC is more efficient than $SO_2$ against yeasts, as it does not induce the yeast cell into a VBNC state, as it happens for $SO_2$, but it leads to cell death [70]. However, its use is recommended in addition to $SO_2$, because DMDC does not have antioxidant activity, and, at approved doses, it is ineffective against bacteria. In order to exert antibacterial activity, high and unapproved doses would be necessary. In fact, doses of 1000 mg/L are necessary to inhibit the growth of *Acetobacter aceti*, whereas doses of 500 mg/L are necessary for the inhibition of *Lactobacillus* spp. [68].

The antimicrobial efficiency, however, depends on the temperature, the pH, the ethanol content, the microbial species, and its initial concentration. Very early results [71] suggested that DMDC was more effective in wine rather than grape must due to the synergistic action with ethanol. Some authors [51] demonstrated that the species most sensitive to DMDC are *Z. bailii*, *Zygoascus hellenicus*, and *Lachancea thermotolerans*, whereas other species, such as *Sch. pombe*, *D. bruxellenisis*, *S. cerevisiae*, and *Pichia guilliermondii*, are also sensitive, but at slightly higher concentrations. More recent studies [72] evaluated the efficacies of the addition of the legal limit of DMDC against different yeasts associated with grapes and wines, confirming the high sensitivity for *Zygosaccharomyces* species, whereas for other species, such as *Candida*, *Metschnikowia*, *Meyerozyma*, or *Wickerhamomyces*, it was demonstrated that the additive may not provide long-term microbiological stability. The response of *Brettanomyces* to DMDC varied, indicating the need for additional research with a high number of strains in order to clarify the efficacy of this additive for long-term microbiological stability.

### 3.4. Chitosan

Chitosan is the main derivative of chitin, a biopolymer composed of N-acetylglucosamine units linked by β (1→4) linkages. This polysaccharide is the most abundant in nature after

cellulose, and it is found mainly in mollusks, crustaceans (the main commercial source), fungi, and insects [73]. Chitin is insoluble in common solvents, whereas chitosan, obtained by the removal of the acetyl group by deacetylation reaction, is a more soluble form [74,75]. Chitosan, indeed, is soluble in weak organic acid solutions at different concentrations, based on its degree of deacetylation (DD) and the MW [76]. These properties (DD and MW) influence the suitability of chitosan for different applications.

Actually, the biological activities of chitosan, including the antimicrobial activity against foodborne, filamentous fungi, yeast, and bacteria, have attracted notable interest for its use as a potential food preservative of natural origin [77].

Recently, in winemaking, chitosan has been accepted by the European Commission (Reg CE 53/2011) as a fining agent for the treatment of wines, for different purposes; in this sector, only chitosan derived from *Aspergillus niger* is admitted, in order to avoid any potential concerns of allergenicity correlated to the crustacean raw material.

The OIV (oiv-eno-338a-2009) has authorized the use of chitosan in different doses for the following applications:

- The addition of 100 g/hL is authorized to prevent hazing in wine, and to reduce the concentration of heavy metals (Fe, Pb, Cd, Cu). Different studies confirmed the clarifying action of chitosan and the prevention of protein haze phenomena, mainly in white wine, with the use at the permitted doses [78–80]. Similar results were found also in matrices different from the wine, such as beer [81] or fruit juices [82,83]. Chitosan also has an action of sorption of heavy metals, such as iron and copper, avoiding the formation of hazing phenomena in wine. A reduction of 70% of iron and 30% of copper is observed at a dose of 1000 mg/L [84], whereas other authors reported the ability to reduce the content of iron, lead, and cadmium in wine by adding doses of this polysaccharide ranging from 200 to 2000 mg/L [85].
- Doses of 500 g/hL are allowed to reduce any contamination by ochratoxin A (OTA), a mycotoxin which can be present in wine at a maximum dose of 2 µg/L (EC 2005), produced by fungi of the genera, *Aspergillus* and *Penicillium*, classified in group 2B as a "possible human carcinogen" by the International Agency for Research on Cancer [86]. Different studies demonstrated the ability of chitosan, at doses of about 4000–5000 mg/L, to remove a large percentage of OTA in wine [87,88].
- Amounts of 10 g/hL can be used to reduce the concentration of unwanted microorganisms, especially *Brettanomyces* spp. Details regarding the antimicrobial activity will be reported below (see Sections 3.4.1–3.4.3).

A further action of chitosan in the winemaking process is the antioxidant activity, although the use for this activity is not yet officially approved by the OIV. Indeed, Chinnici et al. [84] observed that the addition of 1000 mg/L of chitosan inhibits oxidation and, therefore, the browning of wine, with an action comparable to the addition of 80 mg/L of $SO_2$. Other authors [89,90] also showed the anti-radical action exerted by chitosan at doses ranging from 100 to 2000 mg/L. Furthermore, this compound has been the subject of a GRAS (*Generally Recognized as Safe*) notice to the United States Food and Drug Administration (US FDA) for its intended use in wine. However, due to its recent introduction in winemaking, the actual range of applications and the potential limitations of its use have not yet been fully clarified [91]; for example, Scansani et al. [92] reported that the addition of chitosan in fermentations with *S. cerevisiae* and *Sch. pombe* influences the final chemical composition of wine.

### 3.4.1. Antiseptic Activity

The antimicrobial mechanism of action of chitosan is not yet well elucidated; however, several mechanisms have been proposed (Figure 2).

In particular, the most accredited hypothesis is an electrostatic interaction between the positive charges of chitosan molecules and the negative charges of the teichoic acids in the cell wall of the Gram-positive bacteria (Figure 2A) [93–96]. These interactions cause a series of reactions, including a permeability increase, osmotic imbalance, interference with

the electron transport chain, and bacterial cell death. Furthermore, the different microbial cells show different degrees of sensitivity to chitosan, as a consequence of the different cell structure [94]. In fact, the microorganisms that contain chitin in their cell wall (as yeasts) or Gram-negative bacteria (their outer membrane essentially consists of lipopolysaccharides with negatively-charged phosphate and pyrophosphate groups) have been found to be less susceptible to chitosan [75,97].

Other authors [98] proposed that chitosan, once absorbed by the bacterial cell, interacts with its DNA, causing an inhibition of DNA transcription, mRNA synthesis, and, therefore, protein synthesis (Figure 2B).

A further mechanism was proposed by Ralston et al. [99], according to which, chitosan forms a layer on the yeast cell surface that hinders the entry of nutrients (as glucose) into cells, leading to their death (Figure 2C).

Finally, Guibal [100] and Kong et al. [101] observed that chitosan chelates micronutrients and metal ions important for the stability of the outer membrane in Gram-negative bacteria [102], leading to eventual cell death (Figure 2D).

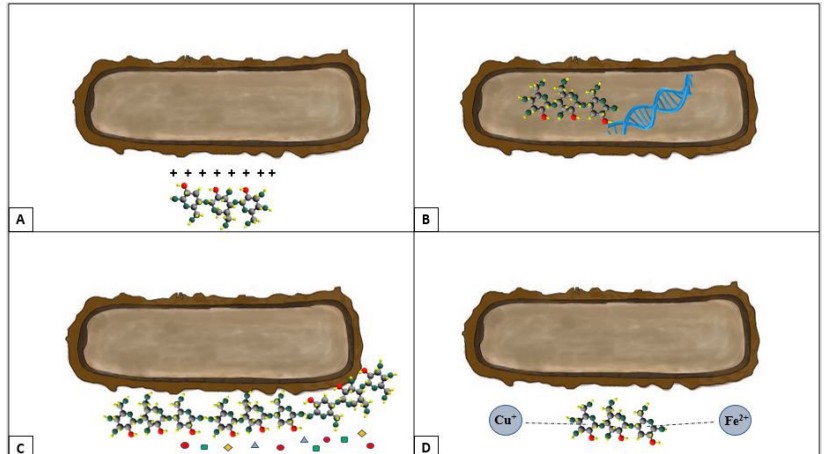

**Figure 2.** Chitosan antimicrobial mechanisms of action. (**A**) = Electrostatic interactions with the cell wall of Gram-positive bacteria [93–96]; (**B**) = Interaction with bacterial DNA [98]; (**C**) = Formation of chitosan layer on the yeast cell surface [99]; (**D**) = Chelation of micronutrients and metal ions with impact on the Gram-negative bacteria cell wall [100–102].

Generally, chitosan has a stronger antimicrobial activity against bacteria rather than against fungi [103]. Furthermore, the antibacterial effects of chitosan are dependent on its MW [104,105] and the DD [103].

### 3.4.2. Activity against Yeasts

In relation to wine yeasts, chitosan generally shows higher inhibitory effects towards non-*Saccharomyces* species than *S. cerevisiae* [52,106], and this aspect is very useful for the correct management of alcoholic fermentation. Indeed, it has been reported that chitosan has a biocidal action on *S. cerevisiae* only at doses higher than levels permitted in winemaking by the OIV [52,97]; only a 2–3 day increase in lag-phase has been observed, with differences correlated to the different *Saccharomyces* strains [107]. The observed differences among *S. cerevisiae* strains might be associated with the content of constitutive polyunsaturated fatty acids in the yeast cell membrane, which is correlated to the permeability and fluidity of the membrane. Strains with higher amounts of these compounds are more susceptible to chitosan, as this compound can enter more easily into their cytoplasm [108,109].

As regards the inhibition of non-*Saccharomyces* yeasts, the effectiveness of chitosan at a dosage of 400 mg/L on the most frequent species present in red grape must is lower than the effect of 50 mg/L $SO_2$ addition [110]. Other authors [111] analyzed the effect of chitosan in apple juice and elderflower against yeasts frequent in grape must. They reported that

a dose of 400 mg/L is able to inactivate *Hanseniaspora uvarum* and *Z. bailii*, and a dose of 300 mg/L is able to inactivate *Candida* spp. and *Rhodotorula* spp.

Among non-*Saccharomyces* species, particular attention should be paid to the effect on the contaminant *Brettanomyces/Dekkera* spp.; it was found that a 40 mg/L dose is able to reduce the *B. bruxellensis* contamination of red wine aged in barrique [112]. These authors reported that if the "batonnage", which favors the contact of the polysaccharide with the whole mass, is applied after the addition of the chitosan, this may favor the recovery of *Brettanomyces* cells, in consequence of the resuspension and oxygen incorporation. According to these authors, the activity of chitosan can be considered fungistatic rather than fungicidal at these concentrations. Petrova et al. [113] reported that the use of 80 mg/L reduced the microbial population by 3-log in 6–8 days after treatment, even if complete eradication was not observed, given the resumption of growth up to $10^5$ CFU/mL at day 68.

Other studies performed on substrates different from wine showed the inactivation and inhibition of *Brettanomyces* spp., although different and higher doses than those authorized by the OIV were tested [52,106,114,115]. *B. bruxellensis* sensitivity to chitosan seems to be not affected by ethanol, whereas growth inhibition depends on the MW of the polysaccharide: chitosan, with low MW (107 KDa), is more effective than medium- and high- MW (310 and 624 KDa, respectively) chitosan [116].

### 3.4.3. Antibacterial Activity

The antibacterial action of chitosan has been investigated by several authors, with application in the food sector, although no direct studies have been carried out on wine or grape must. The addition of 0.3 g/L of chitosan to an apple/elderflower juice reduced the viable cells of LAB, with a positive effect compared to untreated juice. As the bacterial load was not knocked totally down by chitosan, the total bacterial count in treated juice tended to reach the same level of the untreated after storage for 8 days at 7 °C [111]. Furthermore, the different LAB species are differently sensitive to chitosan; indeed, *Lactobacillus* spp. showed a relatively higher resistance to chitosan than *Pediococcus* spp.; *Lactobacillus plantarum* was found to be more resistant than *Lactobacillus hilgardii*, *O. oeni*, and *Pediococcus* spp. [52,53].

Until now, very few data are available on chitosan activity against AAB; Valera et al. [54] reported that 200 mg/L of chitosan can reduce the population of *Acetobacter* spp. of $10^2$ CFU in wine, showing similar results to the addition of 60 mg/L of $SO_2$, indicating that the efficacy of chitosan against AAB is similar to that of sulfites.

### 3.4.4. Sources of Chitosan

As previously reported, only the chitosan derived from *A. niger* is authorized for use in winemaking. Other sources from which the polysaccharide can be obtained are crustaceans and insects [73]. Currently, the main commercial source of chitin and chitosan comprises waste streams from the fishing industry, mainly the exoskeletons of crustaceans [117], since they contain from 15 to 40% of chitin in the exoskeleton [118], and are the major waste product from the marine food industry. However, the supply of crustacean waste is subject to seasonality and begins after spawning in spring [119], and the sustainability of crustacean farming is currently under debate [120]. Furthermore, the use of chitosan from crustaceans is not authorized in winemaking because of potential allergic reactions due to the release of fish protein into the product [79,121]. This supply chain is also characterized by a strong environmental impact, due to the pollution and waste generation during the processing phases. The global market for chitin and chitosan is expected to reach a volume of 282 thousand metric tons and $1.7 billion by 2027 [122,123], intensifying the need for a search of other sources to satisfy the growing market. A promising and sustainable alternative source is represented by insects, although they have not received much attention until now. Unlike crustaceans, insects are not subject to seasonality and can be easily reared, reproducing in laboratory-favorable conditions to their development, obtaining a large quantity due to their high fertility and reproductive rate [124].

In addition, comparing insect farms with traditional livestock and crustacean chains, it is possible to highlight the low environmental impact: they require much lower soil and water, and emit significantly lower levels of ammonia, carbon dioxide, methane, and nitrogen oxide [125–128].

Among insects, the most promising are the bioconverter species, such as *Hermetia illucens*. The larvae can feed on several different organic substrates of animal and vegetable origin [129–131], even decaying by-products and waste, reducing and converting them into larval biomass, rich in proteins, lipids, and chitin [132–134]. The bioconverter insect breedings, used for unconventional waste disposal methodologies and to produce animal feed or novel food production, generate huge amounts of side streams rich in chitin (pupal exoskeleton, called exuviae, and dead adults) that can be exploited to obtain chitosan with a view to a zero-waste circular economy.

Insect-mediated waste management is a growing trend; indeed, *H. illucens* is processed by around 80% of all EU insect-producing companies, and some industries utilize this insect also for high-quality chitin/chitosan production [135].

It is important to underline that the percentage of chitin varies among insect species and within the same species, in relation to different developmental stages. For example, in *Vespa crabro*, the chitin content for larvae, pupae, and adults is 2.2%, 6.2%, and 10.3%, respectively [136], whereas for *H. illucens* larvae, pupal exuviae, and adults, it is 13%, 31%, and 9%, respectively [137]. Moreover, the extraction method (chemical or biological) can affect the amount of chitin extracted from insects and the respective chitosan [137]. These variables also affect the chemical composition of the polymers, which showed a higher DD (around 90%) and a lower MW (lower than 100 KDa) compared to commercial chitosan. For this reason, it is always fundamental to take into consideration the source and the extraction process, in order to obtain chitosan with appropriate characteristics, according to the specific application.

Recent studies have also demonstrated the antimicrobial activity of *H. illucens* chitosan against Gram-positive and -negative bacteria [138]. These chemical and microbiological characteristics would make it optimal for the control of yeasts and bacteria during winemaking. Indeed, as previously reported, the highest antibacterial activity is given by low MW and high DD [116,139].

Moreover, some studies even reported potential insect allergies; to date, insects are not included as a major food allergen by the US FDA [140,141]. For this reason, it is possible to hypothesize that insect chitin and chitosan, which are polysaccharides, could not provoke an adverse reaction, and their usage in winemaking could be safe.

In conclusion, chitosan might be a promising tool for the reduction of sulfites in wine, as it combines antioxidant and antimicrobial properties, and it can be used in different steps of the winemaking process.

Further studies addressed to the use of this sustainable source of chitin and chitosan in order to support/substitute chitosan derived from *A. niger* (the only one authorized for use in winemaking) can add further value to this adjuvant in the winemaking process.

*3.5. Other Substances*

In this paragraph, other substances showing antimicrobial properties will be discussed. Some of them are admitted for applications other than microbiological stabilization, such as polyphenolic compounds, whereas others are innovative chemical additives, for which the use in winemaking is not yet permitted in EU, such as colloidal silver complex.

3.5.1. Phenolic Compounds

In recent years, the addition of phenolic compounds during winemaking as an alternative to $SO_2$ have been evaluated for their antioxidant and antimicrobial activity [13]. Phenolic compounds or polyphenols, naturally occurring in grapes and wines, are chemical additives permitted in enology by the OIV, following the EU legislation (Reg. EC No. 606/2009 and further modifications), when their amount in grapes and wines is too low.

Indeed, the natural concentration of wine phenolic compounds depends on various factors related to the grape (cultivar, time of the harvest, soil, climate, etc.) and enological practices (maceration time and temperature, fermentation with skins and seeds, enzyme addition, pressing, MLF, etc.) [142,143]. The major phenolic compounds present in wine are phenolic acids (hydroxycinnamic and hydroxybenzoic acids), flavonoids, condensed tannins, anthocyanins, and stilbenes (resveratrol). These compounds are responsible for many wines' organoleptic characteristics, such as color and astringency [144]. Moreover, the polyphenols are also associated with health benefits related mainly to cardiovascular and degenerative diseases [145], due to their well-known antioxidant, anticancer, or anti-inflammatory effects [146]. Several studies have demonstrated the role of phenolic compounds in wine for their antimicrobial activity against LAB and pathogenic bacteria [147,148], such as *Staphylococcus aureus*, *Escherichia coli*, *Candida albicans*, *Salmonella enteritidis*, and *Pseudomonas aeruginosa* [149], suggesting their potential use as novel "natural antimicrobial agents" in winemaking. To date, the action mechanism of phenolic antimicrobial activity is not fully clarified. Several authors have reported that phenols increase cytoplasmic membrane permeability with a leak of bacterial intracellular constituents, and they can also alter the composition of fatty acids [150,151]. Some studies reported their inhibition of the synthesis of peptidoglycan, an essential component of the Gram-positive cell wall; furthermore, the inhibition of nucleic acid synthesis and interactions with cellular enzymes were observed [152–154].

In recent years, Garcia-Ruiz et al. [155] reported a comparative study of the potential inhibitory effects of 18 phenolic compounds (among them, hydroxybenzoic acids, hydroxycinnamic acids, phenolic alcohols, stilbenes, flavan-3-ols, and flavonols) on different LAB strains belonging to the species, *O. oeni*, *L. hilgardii*, and *Pediococcus pentosaceus*, isolated from wine. As expected, the results confirmed that the antimicrobial activity of the wine phenolic compounds is influenced by different phenolic chemical structures. In particular, flavonols and stilbenes showed the highest inhibitory effect on tested LAB growth; phenolic acids and their esters had a medium inhibitory effect, and the flavan-3-ols showed the lowest effect on the enological LAB strains studied. However, the LAB strains tested were more sensitive to the phenols in comparison with $SO_2$ and lysozyme. The antimicrobial activity of phenolics depends upon the microbial species and concentration added. For example, it was found that gallic acid and catechin used in concentrations normally present in wines (about 200 µg/mL) stimulate the growth and increase of the *L. hilgardii* population. On the contrary, at doses of 1000 µg/mL, these compounds had an inhibitory effect on bacterial development [156]. In addition, *O. oeni* seems to be more sensitive to phenolic compounds than *L. hilgardii* [152]. Syringaldehyde, one of the less-investigated polyphenols, represents a promising compound as a potential sulfite substitute because of the good inhibition results against spoilage bacteria (LAB and AAB) and non-*Saccharomyces* yeasts at doses of 250 µg/mL, a concentration that is higher than its sensory threshold in wine (50 µg/mL) [157].

In another study, Stivala et al. [153] showed that hydroxycinnamic acids (trans-p-coumaric and trans-caffeic acid) showed high inhibitory activity towards the growth of wine spoilage LAB strains (*L. hilgardii* and *P. pentosaceus*) in a synthetic wine-like medium (SWM), supplemented with 400 mg/L of each of the tested compounds, resulting in alterations of the bacterial cell integrity due to phenol adsorption.

Among these natural substances, resveratrol, belonging to the stilbene family, has been recently evaluated as an interesting and innovative substitute to $SO_2$ in red wines, for its antioxidant and antimicrobial activity against spoilage microorganisms, such as AAB, LAB, and the yeast of the genera, *Dekkera*, *Zygosaccharomyces*, and *Hanseniaspora* [158].

Phenolic compounds are also very abundant in winemaking by-products, such as grape pomace, seeds, and stems. The exploitation of wine production waste, in order to extract phenolic compounds that can be used as natural preservatives in wine, can constitute a sustainable approach based on the valorization of by-products, turning them into an interesting product with added value in the framework of a circular economy [149,154].

In this context, the olive mill waste extract, with a high hydroxytyrosol concentration, obtained from olive mill waste through a patented process [159], is a natural compound, recently tested as a potential alternative to $SO_2$ in winemaking. Hydroxytyrosol is a phenylethyl alcohol, naturally found in wine in a wide concentration range, with high antioxidant and antimicrobial capacity, and among other oil polyphenols, has been recently accepted as a protective compound against oxidative damage [160]. Although it was demonstrated that this olive hydroxytyrosol-enriched extract might be a suitable source of both antioxidants and antimicrobials, giving good results in model wine, the effectiveness of this compound in comparison to $SO_2$ depends on the microorganisms. Its antimicrobial activity was similar to that of $SO_2$ for *H. uvarum*, *Candida stellata*, *L. plantarum*, *Pediococcus damnosus*, and *A. aceti*; higher for *O. oeni*; and lower for *D. bruxellensis* and *Botryotinia fuckeliana*. However, this extract itself is not sufficient for the effective replacement of $SO_2$ in wines, but the well-known health properties and bioavailability of hydroxytyrosol would increase the added value of these wines with low $SO_2$ content. Although good results have been obtained with this compound in model wine, further experiments on real wines should be performed to confirm its usefulness.

### 3.5.2. Colloidal Silver Complex

The antimicrobial properties of silver have been known since ancient times. Silver nanomaterials are used in the food sector for water purification [161], and also in new packaging materials with antimicrobial properties [162]. Recent studies report the antimicrobial activity of these materials against a large scale of Gram-negative and Gram-positive bacteria, other than some antifungal and antiviral activities [163]. Although the exact action mechanism has not been fully elucidated yet, the existing experimental evidence supports different mechanisms correlated to the physicochemical properties of these materials, such as size and surface, which allow them to interact or cross cell walls or membranes, directly affecting intracellular components. The first action mechanism postulates that these particles cross the outer membrane, accumulating in the inner membrane, where the adhesion of the nanoparticles generates cell destabilization and damage, and, subsequently, its death. The second mechanism proposes that nanoparticles can also enter into the cell, where they interact with sulfur or phosphorus groups, present in DNA and proteins, altering their structure and functions. In the same manner, by interacting with thiol groups of the enzymes, they induce the formation of reactive oxygen species and free radicals, generating damage to the intracellular machinery. A third mechanism is the release of silver ions from the nanoparticles, which can interact with cellular components, altering metabolic pathways, membranes, and even genetic material. Colloidal silver complex was tested for its effectiveness as an antimicrobial agent instead of $SO_2$ in both white and red winemaking [164]. The authors found that doses of 1 g/kg of grapes are able to control AAB and LAB development, allowing the growth of *S cerevisiae* was at rates similar to those observed with $SO_2$. The silver concentration in finished white and red wines was 18.4 mg/L and 6.5 mg/L, respectively, which were below the legal limits of 100 mg/L, established by the OIV for silver content in the final wines. Other authors [165,166] reported the effects of two silver nanoparticles coated with biocompatible materials (polyethylene glycol and reduced glutathione) against relevant wine-related microorganisms. These new materials have exhibited great potential to be used as antimicrobials to control LAB and AAB after alcoholic fermentation, even more effectively than $SO_2$, and their action against yeast was greater than that of $SO_2$.

However, the precise effects on different wine-related microorganisms and the conditions for an effective application of these nanomaterials in wines needs further consideration. Furthermore, their toxicity and the impact of this application on the composition and sensory properties of wine have to be evaluated to ensure application at the winery level.

## 4. Conclusions

This review discussed the main chemical compounds with the potential to ensure the microbiological control of the winemaking process maintains the organoleptic properties of the wine as much as possible. Sulphur dioxide is still the main antimicrobial agent used in winemaking to decrease the risk of microbial spoilage, but due to the potential health problems that may arise by its usage, other treatments have been developed and applied to control the activity of undesirable microorganisms in wine.

Therefore, the other chemical compounds currently admitted by the official legislation, such as sorbic acid, lysozyme, DMDC, and chitosan, and other substances showing antimicrobial properties, for which the use for wine microbiological stabilization is not yet permitted in EU, were discussed. To date, a single compound able to completely replace $SO_2$ is not yet available, but a combination of different procedures might be useful to reduce the sulfite content in wine.

Among the proposed alternatives to $SO_2$, the insect-based chitosan might represent an innovation which can meet the consumers' expectations, who are even more interested in the healthy traits of food products and in the use of environmentally friendly practices in the production process. Furthermore, this approach can contribute to increase the competitiveness of wine producers, as the application of pro-environmental business practices may promote the profile of a brand and improve its image.

**Author Contributions:** Conceptualization, A.C., G.S. and F.T.; software, G.S., F.T. and R.P.; resources, F.T., G.S., R.P. and R.S.; data curation, A.C., G.S. and F.T.; writing—original draft preparation, F.T., G.S. and A.C.; writing—review and editing, A.C., P.R., R.P. and C.S.; supervision, P.R., P.F., R.S. and C.S.; funding acquisition, A.C., P.R. and P.F. All authors have read and agreed to the published version of the manuscript.

**Funding:** This research received no external funding.

**Institutional Review Board Statement:** Not applicable.

**Informed Consent Statement:** Not applicable.

**Data Availability Statement:** The data presented in this study are available in the article.

**Acknowledgments:** This work was supported by the projects, PSR Regione Basilicata 2014–2020, sottomisura 16.2 IN.VINI.VE.RI.TA.S (Innovare la viti-VINIcoltura lucana: VErso la RIgenerazione varieTAle, la Selezione di vitigni locali e proprietà antiossidanti dei vini), N. 976, NOBILAPIO"—sottomisura 16.1. Azione 2–PSR Campania 2014/2020–N° H12C19000130009 and PO FESR BASILICATA 2014–2020 "RETREAT" project D.D. 12AF.2020/D.00424. Furthermore, the JRU MIRRI-IT (http://www.mirri-it.it/, accessed on 13 July 2021) is greatly acknowledged for scientific support.

**Conflicts of Interest:** The authors declare no conflict of interest.

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
