# Peer review of "Chemical Methods for Microbiological Control of Winemaking: An Overview of Current and Future Applications"

_beverages, doi:10.3390/beverages8030058_

Round 1

Reviewer 1 Report

The use of SO2 in winemaking as well as efforts to replace it is an age-old topic. Despite a lot of published materials on this field, the satisfactory solution still has not been found. The topic of alternative/supplementary methods to SO2 has become a trend again in recent years. Primarily because of the potential risks to human health as well as due to the requirements to produce wines with minimal intervention and influence on the sensory profile. The title of the review paper promises a practical and up-to-date view of this topic. However, the content is quite brief and does not meet expectations. Further, the paper is rather unbalanced and not exactly objective:

·         Why are potential health risks mentioned for SO2 and not for the other four substances?

·         I understand that these four agents were selected for review due to legislative permission, but they are certainly not novelties. I recommend mentioning at least their traditional use in winemaking/food industry together with the latest studies regarding their application and efficiency.

·         Furthermore, I have reservations about the unjustified emphasis on chitosan. Although the production of this material through an insect farm is very interesting, no data or comparison with the production of other substances is given anywhere in the manuscript.

For the above reasons, I recommend that the manuscript should be better organized and enriched with missing knowledge.

Reviewer 2 Report

In this manuscript, the authors tackle with the topic of the use of various chemical methods as substitutes of SO2 for the microbial control under winemaking processes. As announced by the title, the authors present the discussed topic through the sustainability approach and more precisely, they focus on how those alternative chemical methods can constitute sustainable solutions in the winemaking industry. The authors submitted the draft for reviewing in order to be published in the special issue Role of Microorganisms in Wine Production: From Vine to Wine of the journal Beverages.

The topic is very interesting by itself and the draft is well structured. English language proofreading is required in order to meet the high standards of the journal.

Although the topic sounds relevant with the theme of the under-publication special issue, my main concerns (or objections) after reading the draft are summarized as:

1. Low degree of novelty. The most part of the discussed topic has been already reviewed quite much in previous and recent papers. For example,  

·         Santos et al., 2011, Chemical and physical methodologies for the replacement/reduction of sulfur dioxide use during winemaking: review of their potentialities and limitations, https://doi.org/10.1007/s00217-011-1614-6

·         Castro et al., 2022, Chitosan as an antioxidant alternative to sulphites in oenology: EPR investigation of inhibitory mechanisms,  DOI: 10.1016/j.foodchem.2019.01.155

The part covering the sustainable approach of the use of insect and crustacean -extracted chitin, has a degree of novelty but the sustainability claim is not justified thoroughly. Although the insect production seems to have a lower environmental impact, whether the insect production is sustainable, has not proved yet and it is a current debate among researchers.

Why the use of chitin is sustainable? Is it its use or its extraction processes sustainable in order to justify the sustainability claim?

2. Other than announced and expected by the title, the authors do not offer a holistic review about the sustainable approach of the SO2 alternatives. Which sustainability dimension is discussed, the environmental, the economic, the social, or all of these? This should be addressed with current and state-to-the art reviewed and justified scientific data and reports. As in the conclusion part, the authors refer to ‘’ the use of environmentally friendly practices in the production process’’, this should be supported in the draft by data from e.g., Life Cycle Analyses concerning the environmental impact of those chemicals in terms of e.g., Carbon footprint (see e.g., Ferrara and De Feo, 2018, https://doi.org/10.3390/su10020395).

3. Other innovative approaches on the topic are missing and should be reviewed too e.g., α-pinene (Hou et al., 2019, https://doi.org/10.1080/10942912.2020.1716798), hydroxytyrosol (Raposo et al., 2016, https://doi.org/10.1016/j.lwt.2015.08.005), or even indirect to chemical methods, such as biochemical methods (bio-protection) where the excretion of biomolecules from other biological agents functions as an alternative to SO2

4. The authors do not clearly structure in the draft which winemaking technique they refer each time to. In other words, which chemical SO2 alternatives are exclusively aimed to certain types of wine, to white winemaking, or red one, or both, at which winemaking stage, and under which physicochemical conditions (e.g., pH). This information could be summarized in separate table (see my general comments/proposals). Moreover, the SO2-minimal use or non-use during the production of natural wines, should be discussed too.

5. Imbalance of the presented material. A big part of the drafted focuses on SO2 with information already known from previous studies, while the main theme is the SO2 sustainable alternatives.

Other general comments/proposals

-The SO2 acceptable limits during the various stages and conditions of different winemaking techniques, should be thoroughly presented.

-The limits and the corresponding legislation and winemaking techniques, stages and conditions of the addition in winemaking process of reviewed chemicals, should be summarised in separate table so that the readers can easier compared and understand the topic.

Specific comments

Lines 44- 45: ’’ The control of wild microorganisms present in grape must or wine is an advisable oenological practice to ensure the imposition of starter cultures ’’. I consider it is the control of just grape must before the inoculation with starter cultures, as the wine is not ready yet. If you refer to the malolactic fermentation after the finished wine, this should be clearly shown so that the readers can understand. Please descibe a bit more.

Line 50: ‘’ irreversible effects on wine quality and considerable economic losses’’. Which are those? Please describe so that the readers can understand. 

Lines 55-65: You refer to consumers’ acceptance of SO2 limits and the relevant legislation. Which are those limits? You also refer to ‘environmentally friendly production’. What is the environmental impact of the use of SO2 in winemaking? Please justify with data.  Which are the ‘organoleptic alterations’? Please provide information that is more concrete.

Line 357 : ’’although clouding phenomena may occur due to protein haze’’. I understand what the authors try to say, but the meaning of this phase must be improved and better explained so that the readers can understand.

Line 383: ‘’Dimethyldicarbonate’’. Please, separate the word into two words.

Reviewer 3 Report

Dear authors,

Please considerer the following suggestions/remarks:

- line 90 – “HSO3-“ correct “-“ to superscript position

- line 106 – correct sulfide (S2) to (S2-)

- Line 147 – instead of use the term “antiseptic” (a disinfectant use on live tissue) use the term “disinfectant”

- Lines 191, 286 – remove “etc...”

- line 230 – as the species is here referred for the first time, use the full species name (Acetobacter pasteurianus). The same observation for other species names.

- the term malolactic fermentation was introduced in line 47 – put there the abbreviation (MLF)

- line 281 – “rosé wines” – considerer to change to “rosé or pink wines”

- in table 2 – verify the lysozyme structure. In average the molecular weight of lysozyme is 14/15 kDa which corresponds more or less to 129 amino acids. In the chemical structure is a fraction of that. Instead of the chemical structure, put some characteristics: isoelectric point, molecular weight. Or put a note such as “part of the chemical structure”, although it is not usual to present a protein like this. In chitosan put a space between the admitted amount and its unit.

- line 455 – “of the genus Aspergillus and Penicillium” – of the genera A… and Pe…

- uniform the abbreviation to Schizosaccharomyces pombe: sometimes is Sch., others Schiz.

- line 500 – “microbial cell membranes” – please, rephrase it and avoid generalizations. In some microorganisms affects the membranes, in Gram-positive bacteria may affect the cell wall, namely teichoic acids.

- fig. 2 – In figure caption should be given example of the organisms of each mechanism of action and, eventually references.

- remove the paragraph between lines 536 and 537.

- lines 578/571 – the phrase is confuse: “since shrimps” to “fish protein”. Clarify it.

Round 2

Reviewer 1 Report

I appreciate the additions and expansion of the manuscript and I recommend it for publication with only one small revision, i.e. between lines 51 and 55 the same information is repeated twice, please reduce it:

l. 51-53 “…whereas the control of microorganisms present in the wine is necessary in order to assure or the dominance of suitable yeast/bacteria strains for adequate alcoholic or malo lactic fermentation (MLF)…”

l.54-56 “Furthermore, the control of microbial load in the final product is necessary to avoid wine spoilage due to the growth of undesirable yeasts, lactic acid bacteria (LAB) and 55 acetic acid bacteria (AAB),…”

Reviewer 2 Report

After the first submission of the draft, my opinion - among others- was that the main issue was the lack of novelty, as information that has already been published in previous published reviews is reviewed again in the present manuscript. Although the authors have made revisions, they have not addressed the main inherent weakness of the manuscript, thus the repetition of already known and published information - apart from the chitin part-,  still remains. For this reason, I am regrettably forced to maintain my previous evaluation. 
